# Auxin-sensitive Aux/IAA proteins mediate drought tolerance in Arabidopsis by regulating glucosinolate levels

Mohammad Salehin[1], Baohua Li[2], Michelle Tang[2], Ella Katz[2], Liang Song[3], Joseph R. Ecker [3], Daniel J. Kliebenstein [2] & Mark Estelle[1]

A detailed understanding of abiotic stress tolerance in plants is essential to provide food security in the face of increasingly harsh climatic conditions. Glucosinolates (GLSs) are secondary metabolites found in the Brassicaceae that protect plants from herbivory and pathogen attack. Here we report that in Arabidopsis, aliphatic GLS levels are regulated by the auxin-sensitive Aux/IAA repressors IAA5, IAA6, and IAA19. These proteins act in a transcriptional cascade that maintains expression of GLS levels when plants are exposed to drought conditions. Loss of IAA5/6/19 results in reduced GLS levels and decreased drought tolerance. Further, we show that this phenotype is associated with a defect in stomatal regulation. Application of GLS to the *iaa5,6,19* mutants restores stomatal regulation and normal drought tolerance. GLS action is dependent on the receptor kinase GHR1, suggesting that GLS may signal via reactive oxygen species. These results provide a novel connection between auxin signaling, GLS levels and drought response.

[1] Section of Cell and Developmental Biology and Howard Hughes Medical Institute, University of California, San Diego, La Jolla CA. 92093, USA. [2] Department of Plant Sciences, University of California, Davis, CA 95616, USA. [3] Genomic Analysis Laboratory, Howard Hughes Medical Institute and The Salk Institute for Biological Studies, La Jolla, CA 92037, USA. Correspondence and requests for materials should be addressed to M.E. (email: mestelle@ucsd.edu)

The Aux/IAA proteins are a large family of auxin co-receptors and transcriptional repressors that have a central role in auxin signaling. In the presence of auxin, the Aux/IAAs are degraded through the action of the ubiquitin E3-ligase SCF$^{TIR1/AFB}$, resulting in de-repression of transcription by the AUXIN RESPONSE FACTOR (ARF) transcription factors[1,2]. Although the mechanisms of auxin perception and Aux/IAA degradation are well known, other aspects of Aux/IAA regulation remain poorly understood. In particular, the factors that regulate transcription of the *Aux/IAA* genes are mostly unknown. Transcription of many *Aux/IAAs* genes is auxin regulated via the ARF proteins resulting in robust negative feedback regulation of auxin response. In addition, expression of *SHY2/IAA3* by the ARR1 transcription factor is important for cell differentiation in the root[3]. However, because expression of the 29 *Aux/IAA* genes in Arabidopsis is highly dynamic with both shared and specific expression domains, they are probably regulated by many additional transcription factors. In this way, auxin response pathways can be integrated with other signaling pathways that respond to environmental or genetic signals. As proof of this concept, we recently showed that three *Aux/IAA* genes, *IAA5, IAA6*, and *IAA19* are directly regulated by DREB2A and DREB2B, transcription factors that are known be important for drought response[4]. Further we demonstrated that these *Aux/IAA* genes are required for drought tolerance. Here we show that these Aux/IAAs act to regulate the levels of aliphatic glucosinolates (GLS), well characterized secondary metabolites that contribute to herbivore tolerance and innate immunity. We also demonstrate that GLS compounds have an important role in stomatal regulation and drought tolerance.

## Results

**IAA5/6/19 regulate glucosinolate biosynthesis**. To determine the molecular basis of reduced drought tolerance in *iaa5, iaa6*, and *iaa19* mutant plants, we used RNAseq to identify genes that were differentially regulated in Col-0 vs. the *iaa5 iaa6 iaa19* (*iaa5,6,19*) triple mutant when exposed to desiccation stress (Supplementary Data 1). A total of 651 genes were differentially expressed between the mutant and Col-0 under these conditions (FDR < 0.001), 321 down-regulated and 330 up-regulated. A gene ontology search revealed that 14 genes that function in the aliphatic glucosinolate (GLS) biosynthetic pathway are down-regulated in *iaa5,6,19* (Fig. 1a, b). In contrast, expression of these genes is not significantly affected by dehydration stress in Col-0. We confirmed some of these results by quantitative RT-PCR (qRT-PCR) (Supplementary Fig. 1a).

Because the aliphatic GLS biosynthetic enzymes are down-regulated in *iaa5,6,19* mutants during drought stress, we wondered if the levels of GLSs were also affected. To test this, we measured GLS levels in stress treated Col-0 and *iaa5,6,19* mutants at time intervals. Indolic GLSs were unaltered (Supplementary Table 1). However, the level of 4-methylsulfinyl glucosinolate (4-MSO), the most abundant aliphatic glucosinolate in Arabidopsis (Col-0), was sharply decreased in *iaa5, 6, 19* plants after 1 h and 3 h of desiccation (Fig. 1c). These data shows that down-regulation of aliphatic GLS biosynthetic enzymes in *iaa5, 6, 19* mutants results in decreased GLS levels.

**Aliphatic GLSs are required for drought tolerance**. GLSs are well known for their role in plant defense and innate immunity, although recent studies also suggest that they may have a role in regulating plant growth[5–9]. These secondary metabolites are found primarily in the Brassicaceae, a family that includes many economically important crops[7]. GLSs are broken down by the enzyme myrosinase into thiohydroximate-O-sulfonates which

rearrange to form diverse isothiocyanates (ITC), nitriles and related compounds[8–12]. To determine if decreased drought tolerance in the *iaa5,6,19* mutant is related to reduced GLS levels, we measured the effects of mutations in GLS biosynthetic genes on response to drought. We employed two assays; growth of seedlings on agar medium containing PEG, and growth of plants in pots after withholding water. We first determined the response of the *iaa5,6,19* mutant to water withholding. The results in Fig. 2a, b) confirm that the mutant is less tolerant to drought conditions. Next we characterized mutants in the *CYP79F1* and *CYP79F2* genes, encoding enzymes that convert elongated methionine to aldoximes (Fig. 1a). We tested the *cyp79f1f2* double mutant and found that it was less tolerant to both PEG treatment and water withholding (Fig. 2c, d; Supplementary Fig. 2a). Similarly, loss of CYP83A1, responsible for conversion of aldoximes to *aci*-Nitro compounds, results in reduced tolerance to water withholding in pots (Supplementary Fig. 2b). These results indicate that loss of aliphatic GLS compounds results in decreased drought tolerance, providing an explanation for the phenotype of the *iaa5,6,19* mutant.

The MYB28 and MYB29 transcription factors are known to regulate genes in the aliphatic GLS biosynthetic pathway, including the *CYP79F1/F2* and *CYP83A1* genes[13–16]. Indeed, aliphatic GLS levels in the *myb28 myb29* double mutant are extremely low, whereas overexpression of either *MYB28* or *MYB29* results in elevated aliphatic GLS levels[14–17]. Thus, it is possible that the effects of the *iaa5,6,19* mutations on the pathway are mediated by changes in expression of these transcription factor genes. Examination of our RNAseq data revealed that *MYB28* is down-regulated in the triple mutant in response to stress. *MYB29* is expressed at a very low level in seedlings. We confirmed this by qRT-PCR (Supplementary Fig. 1). When we determined the response of the *myb28 mby29* double mutant to water withholding we found that it is less tolerant than the wild type, further support for the idea that GLSs are required for drought tolerance (Fig. 2e; Supplementary Fig. 3a). Since overexpression of *MYB28* and *MYB29* increases GLS levels, we also tested the behavior of these lines in our assays. Strikingly, we found that both lines displayed strongly increased drought tolerance (Fig. 2f, Supplementary Fig. 3b) further confirming that GLSs confer drought tolerance.

Although the role of MYB28/29 in the regulation of GLS biosynthesis is well known, it is possibly that these transcription factors regulate other genes that contribute to drought tolerance. To address this possibility, we also tested the effect of overexpression of the *AOP2* gene on drought tolerance. AOP2 converts 4-MSO to but-3-enyl glucosinolate. Arabidopsis Col-0 lacks AOP2 but overexpression of *Brassica oleracea AOP2* in Col-0 results in an increase in aliphatic GLS levels, albeit less than in lines over-expressing *MYB28*[18]. The results in Fig. 2f and Supplementary Fig. 3b show that increased AOP2 levels results in a modest but statistically significant increase in drought tolerance confirming that MYB28/29 overexpression affects drought tolerance by increasing GLS levels.

If decreased drought tolerance in *iaa5,6,19* plants is due to a GLS deficiency then overexpression of *MYB28* or *MYB29* in the triple mutant may ameliorate the effects of the mutations. Indeed, when we cross the *35 S:MYB28* or *35 S:MYB29* transgene into the *iaa19-1* mutant the result is the restoration of wild-type levels of drought tolerance to the mutant line (Fig. 2g).

**WRKY63 regulates *MYB28/29* expression**. Although IAA5, 6, and 19 are required for expression of *MYB28*, they are unlikely to directly regulate *MYB28* because they are transcriptional repressors[1]. To identify transcription factors that might be direct targets

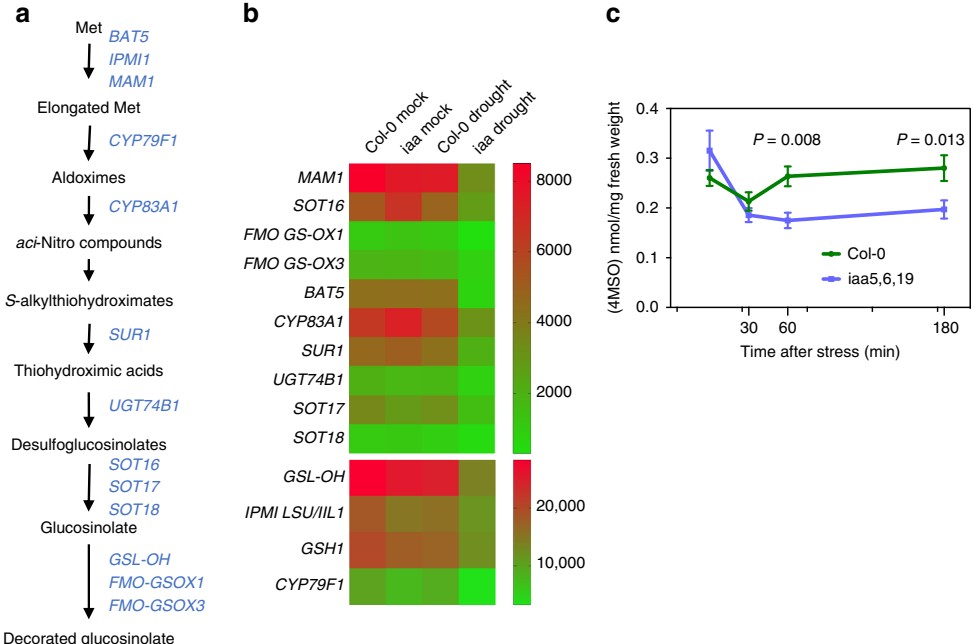

**Fig. 1** The *iaa5,6,19* mutant has lower aliphatic glucosinolate levels after dehydration stress. **a** Aliphatic glucosinolate biosynthetic pathway. The genes listed to the right are all down-regulated in the triple mutant compared to the wild type after dehydration. **b** Heat map showing that aliphatic glucosinolate biosynthetic genes are down regulated in the *iaa5,6,19* mutant during desiccation stress. **c** Time course measurement of 4-MSO levels in 7-day-old seedlings of indicated genotypes at time intervals after application of dehydration stress. Results are presented as the means ± SE of one experiment with 6 biological replicates consisting of $n = 20$ pooled seedlings in each. Two independent experiments showed similar a trend. Statistical significance was determined by (Student's *t*-test). $P = 0.008$ and 0.013 respectively for *iaa5619* at 60 and 180 min of desiccation. Source data are provided as a Source Data file

of the Aux/IAAs, and that might regulate *MYB28* expression, we searched our RNAseq data for factors that are up-regulated in the triple mutant in response to stress. One interesting candidate was *WRKY63*, also known as *ABA OVERLY SENSITIVE3* (*abo3*)[19]. The *abo3* mutant was originally isolated because it is hypersensitive to ABA in seedlings. Our qRT-PCR experiments confirmed that *WRKY63* is up-regulated in response to dehydration in the triple mutant compared to the wild type (Fig. 3a). Examination of the promoter region of the *WRKY63* gene revealed the presence of two tandemly repeated *AuxRE* elements (−350 to −361), known to bind ARF transcription factors[2]. To determine if IAA19 binds to these sequences, we performed a ChIP-PCR analysis using a *rAA19-YPet-His-FLAG* line. This line had previously been shown to rescue the *iaa19* mutant[4]. The results, shown in Fig. 3b, show that recovery of *AuxRE* sequences is significantly enriched in the IP from *rIAA19-YPet-His-FLAG* compared to the control indicating that IAA19 binds to this sequence, presumably indirectly through an interaction with an ARF transcription factor.

To determine if WRKY63 might regulate *MYB28/29* expression, we measured RNA levels in two *35 S:WRKY63* lines by qRT-PCR and found that expression of both genes was reduced suggesting that WRKY63 acts to repress expression of the *MYB* genes (Fig. 3c)[20]. Finally, we measured drought tolerance of the *35 S:WRKY63* lines as well as the knockout mutant *wrky63-1*. As shown in Fig. 3d and Supplementary Fig. 3b, the mutant has a normal response to water withholding. However, both overexpression lines are less drought tolerant consistent with reduced expression of *MYB28/29*. We note that *WRKY63* is a member of small clade of 4 genes. It is possible that these genes have an overlapping function in regulation of *MYB28/29*. Our analysis of WRKY63 differs from the earlier work showing that the *abo1* mutant is less drought tolerance[19]. The reason for this discrepancy is unclear.

**GLS compounds regulate stomatal aperture**. In considering how aliphatic GLSs may regulate drought response, we noted that these compounds have been reported to promote stomatal closure[21]. In addition, the myrosinase TGG1 is one of the most abundant proteins in Arabidopsis guard cells. Experiments with the *tgg1 tgg2* mutant that lacks canonical myrosinases indicate that these enzymes are required for the effect of GLS on stomata[21]. Further, ITC, a GLS metabolite, closes stomata in Arabidopsis and *Vicia faba*[22–24]. Together these results indicate that myrosinase-dependent breakdown products of applied GLS promote stomatal closure. To determine if a defect in stomatal regulation might be responsible for reduced drought tolerance in the *iaa5,6,19* and *myb28 myb29* lines, we first examined the stomatal response to drought in epidermal peels. As expected, the stomata on Col-0 plants close in response to drought conditions (Fig. 4a). In contrast, the stomata on both mutant lines failed to respond. We next asked whether application of 4-MSO would promote stomatal closure in wild-type and mutant plants. As a control we also applied abscisic acid (ABA) The results shown in Fig. 4b demonstrate that all three genotypes respond to both ABA and 4-MSO. We further tested the effects of 4-MSO on light-induced opening of stomata from dark-adapted plants. The results in Fig. 4c show that 4-MSO inhibits this response in Col-0 and *iaa5,6,19* plants. These results suggest that the primary basis for reduced drought tolerance in the *iaa5,6,19* line is failure to close stomata in drought conditions. To test this possibility, we applied 4-MSO to wild-type and mutant plants subjected to water withholding in pots. The results in Fig. 2a, b show that application of this GLS restores drought tolerance in the mutant to wild-type levels. To confirm that myrosinase is required for drought tolerance, we examined the response of the *tgg1 tgg2* double mutant to PEG. The results show that the mutant line is less tolerant than wild type

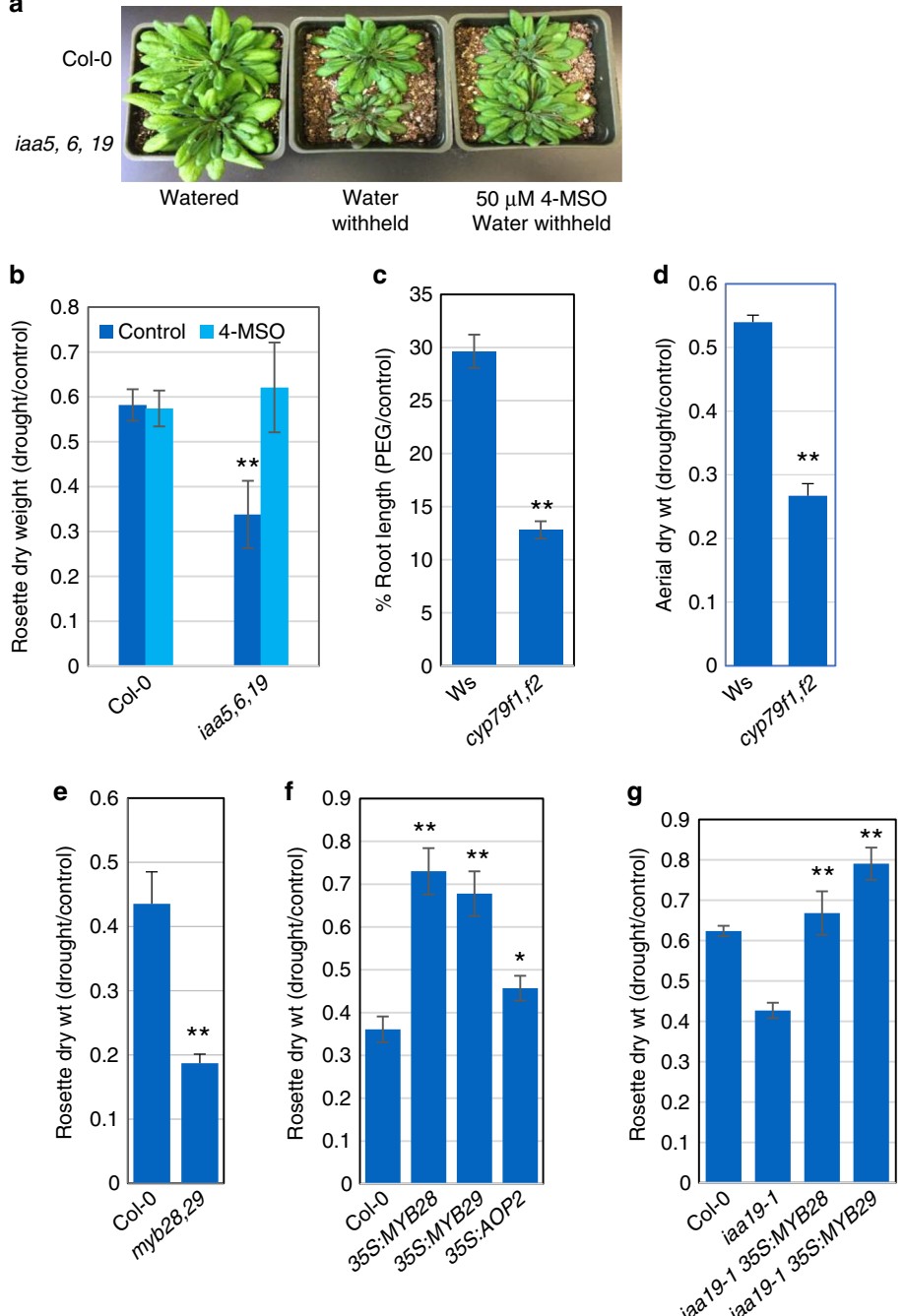

**Fig. 2** Aliphatic glucosinolate deficiency is linked to decreased drought tolerance. **a** The *iaa5,6,19* triple mutant is less drought tolerant than the wild type in a water withholding experiment. **b** Quantification of rosette dry weight from **a**. **c** *cyp79f1,f2* double mutants are sensitive to growth on medium containing PEG compared to the wild-type Ws line. Results are presented as the mean ± SE of one experiments with $n = 12$–15 independent seedlings. **d** The *cyp79f1,f2* line is less drought tolerant than Ws upon water withholding. **e** The *myb28,29* double mutant is less drought tolerant compared to the wild type. **f** 35 S: MYB28, 35 S:MYB29 and 35 S:AOP2 plants display increase drought tolerance. **g** Overexpression of *MYB28* or *MYB29* restores drought tolerance to *iaa19-1* plants. Results are presented as the means ± SE of one experiment with $n = 10$ independent pots. Each experiment was repeated at least twice. For **b**–**g**, differences between mutants and Col-0 or Ws are significant at $p < 0.05$ (∗) and $p < 0.01$ (∗∗) by two-tailed Student's $t$ test. Source data are provided as a Source Data file

confirming that it is a GLS metabolite that contributes to PEG tolerance (Supplementary Fig. 3c).

To determine if all GLS compounds promote stomatal closure, we applied indol-3-ylmethylglucosinolate (I3M), an indolic GLS that is abundant in Arabidopsis, to Col-0. The results in Fig. 4c demonstrate that I3M is unable to inhibit light-induced stomatal opening in either Col-0 or the triple mutant, indicating that not

all glucosinolates have the same efficacy. Similar results were obtained in an experiment where we tested the ability of I3M to promote stomatal closure in light grown plants (Fig. 4d). We also tested a second aliphatic GLS, sinigrin hydrate (SH), that is abundant in many Arabidopsis ecotypes, although not Col-0[25]. Like 4-MSO, SH promoted closure in Col-0 and in the *abi1-2* mutant (Fig. 4e). Thus, it is possible that this activity is restricted

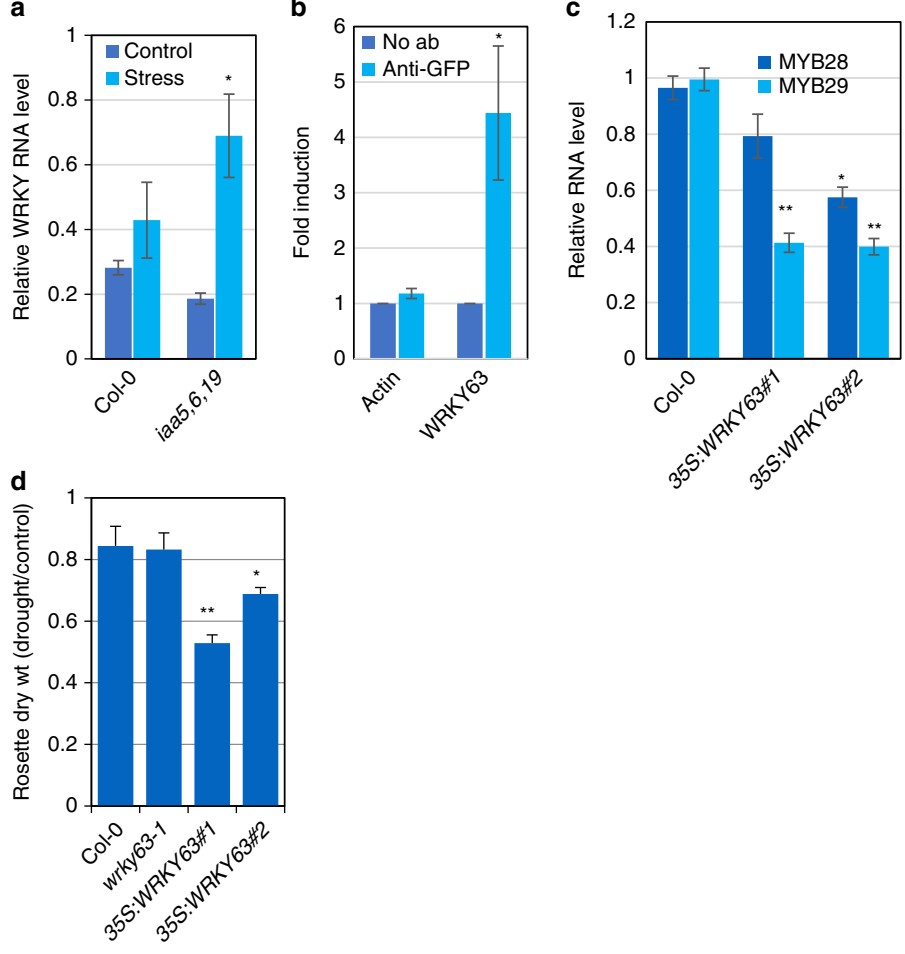

**Fig. 3** Drought activates a transcriptional cascade to regulate GLS biosynthetic genes. **a** Relative *WRKY63* RNA level in the *iaa5,6,19* mutants compared to wild-type seedlings before and after desiccation for one hour. **b** IAA19 binds to a tandem repeat of *AuxRE* elements in the promoter of *WRKY63* (−350 to −361). ChIP was from the *rAA19-YPet-His-FLAG* line. Results are presented as the means ± SE of 3 independent biological experiments with 3 technical replicates in each. Statistical significance was determined by two-tailed Student's t test. **P < 0.05. **c** Relative *MYB28* and *MYB29* RNA level in the seedlings of *35 S:WRKY63* lines. **d** *35 S:WRKY63* lines are less tolerant to drought than wild-type plants after water withholding in pots. Results are presented as the means ± SE of one independent experiments with *n* = 10 independent pots/plants. Differences are significant at *p* < 0.05 (∗) and *p* < 0.01 (∗∗) by two-tailed Student's *t* test. Source data are provided as a Source Data file

to aliphatic GLS compounds, although additional studies are required to explore this possibility further.

**GLS action requires the receptor kinase GHR1.** ABA is known to play a key role in stomatal regulation[26]. We investigated potential interactions between GLS and ABA by testing the response of *iaa5,6,19* to ABA treatment. The results in Fig. 4b show that the mutant line does respond to ABA, indicating that GLS is not required for the ABA response. We also used the *abi1-2* mutant to address the role of ABA in the GSL response. ABI1 is a protein phosphatase 2 C that functions in ABA signaling. The dominant *abi1-1* mutant, which is deficient in ABA regulation of stomatal closure, has a normal response to GLS (Fig. 4e)[26]. These results suggest that GLS and ABA can act independently to regulate stomata. This is consistent with a recent study showing that GLS and ABA have an additive effect on stomatal closure[27].

Reactive oxygen species (ROS) play an important role in stomatal regulation. One of the effects of ABA is to stimulate production of extracellular ROS through the RESPIRATORY BURST OXIDASE HOMOLOG D (RBOHD)[26]. Extracellular ROS than acts through the receptor kinase GUARD CELL HYDROGEN PEROXIDE-RESISTANT 1 (GHR1) to regulate the

SLOW ANION ACTIVATING CHANNEL 1[26]. Since ITCs, products of GLS metabolism, are known to promote stomatal closure via ROS, we wondered if GHR1 is required for response to GLS[23]. The results in Fig. 4f show that the *grh1* mutant is resistant to the effects of 4-MSO on stomata indicating that GLS likely acts through production of ROS.

**Loss of *TIR1/AFB* proteins increases drought tolerance.** Since loss of the auxin-sensitive repressors IAA5, IAA6, and IAA19 in the *iaa5,6,19* results in decreased expression of GLS biosynthetic genes, we predict that the reduction in the levels of these proteins after auxin treatment would have the same effect. Indeed, treatment of seedlings with 10 μM IAA for 2 h results in reduced expression of the GLS genes (Supplementary Fig. 3d). Further, we predict that mutations which stabilize the Aux/IAA proteins will increase both GLS levels and drought tolerance. The TIR1/AFB family of auxin co-receptors consists of 6 members in Arabidopsis that act in an overlapping fashion to regulate auxin-dependent transcription throughout the plant. To determine the role of the TIR1/AFBs in GLS regulation we measured GLS levels in two higher order *tir1/afb* lines, *afb1,3,4,5* and *afb2,3,4,5*[28]. Both lines had significantly higher GLS levels than either Col-0 or

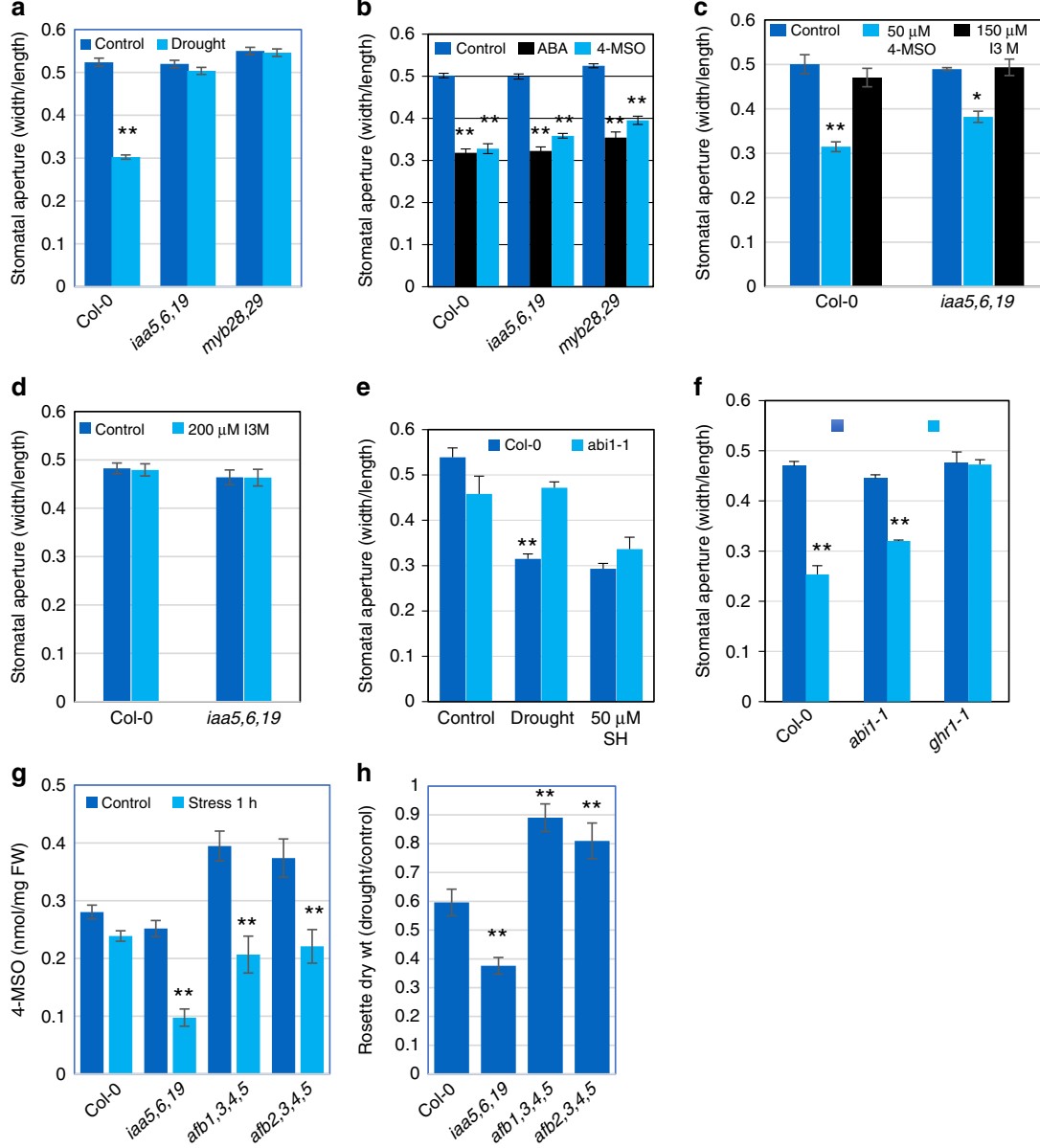

**Fig. 4** Defects in stomatal regulation are responsible for decreased drought tolerance in glucosinolate deficient mutants. **a** Stomatal response to drought in Col-0, *iaa5,6,19* and *mybn28,29* lines. **b** Stomatal response to application of ABA and 4-MSO in Col-0, *iaa5,6,19* and *myb28,29*. **c** 4-MSO inhibits light-induced opening of stomata from dark-adapted plants while the indolic GLS I3M is ineffective. **d** Stomatal response to I3M in light-grown plants. **e** Response of the *abi1-2* mutant to the aliphatic GLS sinigrin hydrate. **f** Response of the *ghr1* mutant to 4-MSO, with *abi1-2* as a control. Unlike Col-0 or *aib1-2*, *ghr1* does not respond to 50 μM SH. **g** 4-MSO levels in *tir1/afb* mutants after 60 min desiccation treatment. **h** The *tir1/afb* mutants display increased tolerance to water withholding in pots. For stomatal closure experiments, $n = 250–300$ stomata from 6 different leaves from 3 different plants grown in the same chamber and growth condidtions. Results are presented as the means ± SE. Differences are significant at $p < 0.05$ (∗) and $p < 0.01$ (∗∗) by two-tailed Student's *t* test. Source data are provided as a Source Data file

*iaa5,6,19* in 7-day-old seedlings (Fig. 4g). After one hour of desiccation, GLS levels dropped in all three mutant genotypes. In the case of the *iaa5,6,19* line, GLS levels were much lower than Col-0, as observed previously. In contrast, GLS levels in *afb1,3,4,5* and *afb2,3,4,5* were similar to Col-0. We also determined the effects of water withholding on these genotypes. The results in Fig. 4h and Supplementary Fig. 3e show that both *afb* lines are significantly more drought tolerant than Col-0.

## Discussion

Stomatal regulation is a complex process that requires the integration of multiple environmental inputs and signaling molecules,

most notably ABA. Here we show that GLS compounds act as a recently evolved signal that regulates stomatal aperture. In Fig. 5, we present a model that can explain our results. In Arabidopsis, auxin acts independently of ABA to regulate stomatal aperture through its effect on GLS levels. GLS degradation products, perhaps ITCs but possibly other compounds, stimulate the formation of ROS. This is consistent with our data showing that the GHR1 receptor kinase is required for the GLS response. However, other scenarios are possible, and several important questions remain unanswered. For example, the source of the GLS and the site of myrosinase action are unknown. Although guard cells are known to have high levels of myrosinase, we have shown that GHR1 is required for the GLS response and it is extracellular ROS

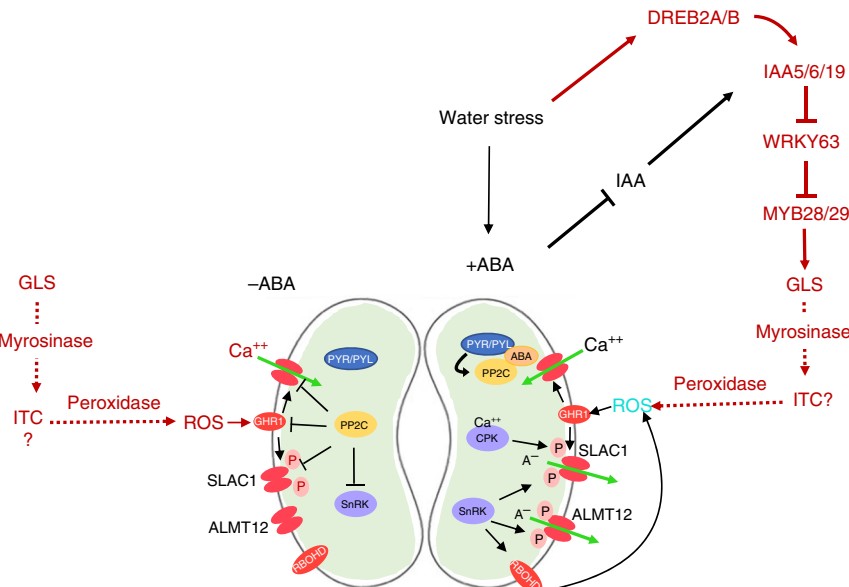

**Fig. 5** A proposed model for GLS action during stomatal regulation. Adapted from[26]. GLS regulation diagrammed in red. Water deficit induces DREB2A/B expression which promotes expression of IAA5/6/19 and maintenance of GLS levels. Myrosinase acts on GLS to produce degradation products, including isothiocyanates (ITC), that promote ROS production via a peroxidase[22]. An increase in ABA levels also results in decreased IAA levels in guard cells[27] which acts to stabilize IAA5/6/19. In the absence of water stress, GLS acts to promote stomatal closure via ROS production

that activates GHR1. Further, the precise identity of the active breakdown product is unknown. These and other important questions await further study.

Although GLS compounds are restricted to the Brassicaceae, allyl ITC has been shown to promote stomatal closure in *Vicia faba* via ROS, suggesting that some components of the GLS signaling pathway may be conserved[24]. Whether or not there is a source of endogenous ITC or related molecule in beans and other non-Brassicaceae remains to be determined. Regardless, our results suggest that other secondary metabolites, in addition to the classical plant hormones, have probably been recruited to perform signaling functions.

Why has auxin, a hormone typically associated with growth regulation, been recruited to regulate GLS biosynthesis? Many recent studies have described the connection between growth regulation and abiotic stress response. For example, gibberellic acid is known to play a key role in regulating plant growth in response to stress[29]. Although there is less information on the role of auxin in growth retardation during stress, we have shown that desiccation or osmotic treatment results in a rapid decrease in auxin response in seedlings, an effect that probably results in decreased growth[4]. By utilizing auxin to regulate stomatal aperture, the plant may integrate growth and stomatal regulation in drought conditions.

## Methods

**Plant materials and growth conditions**. *Arabidopsis thaliana* seed were sterilized with chlorine fumes generated by mixing 49 ml bleach and 1 ml hydrochloric acid. Sterilized seeds were sown on ½ strength MS media, stratified at 4 °C for 2–3 days, and grown in long day conditions (16 h light: 8 h dark) at 22 °C with a light intensity of ~110–130 µmol m$^{-2}$ s$^{-1}$. In addition to wild type Col-0 and Ws lines, the following mutant lines were used as described: *iaa19-1* and *iaa5 iaa6 iaa19*, *dreb2a dreb2b* (ref. [4]); *cyp79f1 cyp79f2* (ref. [4]); *myb28 myb29*, *35 S:MYB28*, *35 S: MYB29* (ref. [17] *S:WRKY63* OE (ref. [20]). *abi1-2* (ref. [30]); *ghr1* (ref. [26]).

**Plasmid construction**. rIAA19-YPet-His-Flag was constructed by recombineering a 1xYPet-6xHis-3xFlag tag to the C-terminus of the IAA19 gene in the Arabidopsis TAC clone JAtY59D10 (refs. [4,31]).

**Transgenic plants**. After floral dip of wild-type Col-0 plants[32], single-insertional transgenic lines of rIAA19-YPet-His-Flag were selected by chi-square test from T2

plants on 1x Linsmaier & Skoog (Caisson Labs, UT) plates containing 15 mg/ml glufosinate ammonium. The expression of the tagged TFs was confirmed by Western blotting. Homozygous transgenic lines were selected from the subsequent generation for bulking seeds.

**RNAseq**. Col-0 and *iaa5,6,19* seedlings were grown on ½ MS for 7 days and desiccated for 1 h on parafilm at room temperature. Total RNA was extracted from three independent biological replicates of each genotype using RNeasy Plant Mini Kits (Cat#79254, Qiagen, CA) and genomic DNA contamination was removed by digestion with Turbo DNase (Cat# AM1907, BioRad, CA) and/or RNase-Free DNase (Cat#79254, Qiagen). RNA quantity was checked by Bioanalyzer for quality control. Library construction and sequencing were performed at the IGM genomics center at UCSD. About 40 million reads were obtained for each sample. Raw reads were processed at Bioinformatics core facility, UCSD School of Medicine using Array Studio software. Alignments were performed using OSA4 and differential expression determined using DESeq2[33]. Gene ontology analysis was performed using Panther[34,35,36].

**ChIP qPCR**. For ChIP qPCR Col-0, *recIAA19: YPet-His-FLAG* and *gWRKY63: WRKY63: eYFP* plants were grown on ½ MS for 7 days in the dark. Approximately 1 g tissues were collected and crosslinked with Pierce™ Methanol-free 16% Formaldehyde (w/v) (Cat#28906, Thermo Fischer Scientific, Carlsbad, CA) diluted to 1%, for 20 min and quenched with 2 M glycine for 5 min. Samples were flash frozen and ground in liquid nitrogen. Chromatin was prepared using EpiQuik Plant ChIP Kit (Cat# P-2014-48, NY, USA) as per manufacturer's instructions. Chromatin was sheared at 4 °C by 16 cycles of 30 s on and 90 s off, on high settings using Bioruptor Plus sonicator device (Cat# B01020002, Diagenode, US). IP was done with 20 µg (10 µL, 2 µg/µL of anti-GFP (#A11122, Thermo Fisher, Carlsbad) at 4 °C overnight. Purified DNA was used for qRT-PCR using SYBR green dye in a CFX96 Real-Time System (Bio-Rad, Hercules, CA).

**qRT-PCR**. Total RNA was extracted using RNA RNeasy Plant Mini Kits (Cat#79254, Qiagen) and genomic DNA contamination was removed by digestion with Turbo DNase (Cat# AM1907, BioRad, CA) and/or RNase-Free DNase (Cat#79254, Quiagen). cDNA was synthesized using SuperScript III (Cat# 18080085, Life Technologies/Thermo Fischer, Carlsbad) and analyzed by quantitative PCR using PowerUp™ SYBR® Green Master Mix (Cat#A25742, Thermo Fischer Scientific, Carlsbad, CA) on a CFX96 Real-Time System (Biorad, Hercules, CA). RNA levels were normalized against PP2A gene. Fold induction was calculated by ddCt method[37]. RNA levels were quantified using the Thermo Scientific NanoDrop 2000[38]. All qRT-PCR experiments were repeated three times. The oligos used for PCR are listed in Supplementary Table 2.

**Glucosinolate Profiling**. Blind experiments were conducted with 3 different genotypes as described above. For each assay 2 independent experiments with 6 independent biological replicates grown under same condition were used.

Typically, 20–30 mg (fresh weight) of pooled 7 day-post-germination seedlings (n = 15–20) were sampled, freeze dried and sent to UC Davis at room temperature for GLS quantification. In total 400 μl 90% (v/v) methanol was added to each sample tube and stored at −20 °C before extraction. The tissues were disrupted with two 2.3 mm metal ball bearings in a paint shaker at room temperature and incubated at room temperature for 1 h. Tissues were pelleted by centrifugation for 15 min at 2500 g and the supernatant was used for anion exchange chromatography in 96-well filter plates. After methanol and water washing steps, the columns were incubated with sulfatase solution overnight. Desulfo-GLS were eluted and analyzed by HPLC[39].

**Drought stress assays**. Osmotic stress assays were performed as described in ref. [4]. Briefly 35 ml ½ MS media with 1% sucrose and 0.8% Bactoagar was autoclaved and solidified in 120 mm square plates and perfused overnight with 50 ml 30% filter sterilized PEG-8000 solution. PEG was drained the next day. 5–6 day-post-germination seedlings were transferred to either ½-MS or ½-MS + PEG plates and incubated at 22 °C in long days. Root length and plant fresh weight were measured after 8 to 10 days.

Drought stress assays in soil were performed as described[40,41] with minor modifications. 7 to 10-day-old seedlings were planted in sectors (two plants per sector) of 8 cm × 8 cm × 10 cm (L × W × H) plastic pots. Plants were grown in a short-day chamber (8 h/16 h. light/dark) at 21 °C during day 18 °C during night with light intensity of 110–130 μmol m$^{-2}$ s$^{-1}$. Seedlings were fertilized regularly. After 18–20 days, pots were saturated with water, drained and weighed. Subsequently, water was withheld for 18–22 days to reduce weight up to 50% and each pot was re-watered to 75% level by injecting 40–50 ml of water in the middle of the pot with a 22-gauge needle. Pots were then subjected to re-drought for 12–15 days to reduce pot weight to 50–60%. Fresh weight and dry-weights of each rosette were measured. For dry weight rosettes were dried overnight at 65 °C in a hot air oven. Drought stress assays were repeated three times with similar results.

In the experiments to determine the effects of GLS on drought tolerance, plants were treated as above except that water withholding began after two weeks of watering. At the same time, plants were sprayed with 50 μM 4-MSO, 50 μM sinigrin hydrate or water 6 times during the first over the next 10 to 12 days. The 4-MSO and sinigrin were dissolved in water.

**Stomata aperture measurement**. Stomatal aperture experiments were performed using fully expanded young leaves from wild-type and mutant plants grown for 3–4 weeks in a growth room with 8 h of light/16 h of dark. To determine the effects of GLSs and drought on stomatal closure, control plants and plants that were subjected to water withdrawal for 3–4 days were placed in the light overnight to open the stomata. Leaves were then incubated in closing solution (5 mM MES-KOH [pH 6.15], 20 mM KCl, and 1 mM CaCl$_2$) under light (110–130 mmol/m2/s) for 2.5 h followed by a further 2.5 h in the presence of 50 μM 4-MSO (Glucoraphanin, Cat# 0009445, Cayman Chemicals), 200 μM I3M (Cat# 2503 S, Alkemist Labs), or 20 μM ABA. I3M was dissolved in DMSO. Control solutions contained comparable amounts of ethanol (ABA) or DMSO (I3M). Abaxial leaf epidermal peels were fixed on glass slide with coverslips[21,42] using Hollister Medical Adhesive Spray. Stomatal apertures were measured with a light microscope (Nikon, Japan). The epidermal peels were examined under a ×40 objective using the microscope. After image acquisition, widths and lengths of stomatal apertures were measured using the ImageJ software (NIH, Bethesda, US). Six independent biological replicates (40–50 stomata from one seedling per replicate) were used for one experiment and at least three independent plants were used.

To determine the effects of GLS on stomatal opening, leaves were incubated in opening solution (10 mM MES-KOH [pH 6.15], 10 mM KCl, and 10 mM iminodiacetic acid) in the dark for 2.5 h with their adaxial surface upward. Upon transfer to the light, 50uM 4-MSO, 50 μM sinigrin hydrate (Cat# S1647, Sigma-Aldrich) 200 μM I3M or 20 μM ABA were added to the solution. Stomatal aperture was measured as above.

**Statistical analysis**. GraphPad Prism 5.0 (San Diego, CA) and Microsoft Excel 2008 were utilized for preparing graphs and statistical analysis. ANOVA utilized for GLS analysis. For others two tailed t-tests were employed.

## Data availability

Plant lines will be available through the Arabidopsis Stock Center. RNAseq data have been deposited into NCBI as GEO#PRJNA549546 (https://www.ncbi.nlm.nih.gov/bioproject/PRJNA549456/). The source data underlying Figs. 2b–g, 3a–d, 4a–h, Supplementary Figs. 1a-f, 2b, and 3c, d are available as a source data file. The authors confirm that any other data supporting findings of this manuscript are available from the corresponding author upon request.

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

## Acknowledgements
We thank Jiyoung Park, Po-Kai Hsu, and Julian Schroeder for *abi1–21* and *ghr1* seeds and for help with stomatal aperture measurement, Eric Schmelz for assistance with freeze drying of samples, Jeongim Kim and Clint Chapple for *cyp83a1* seeds, Venkatesan Sundaresan and Titima Tantikanjana for *cyp79f* single and double mutant lines, and Jim Whelan for *wrky63-1* and *35 S::WRKY63* seeds. This work was supported by grants from the NIH (GM43644 to M.E.; J.R.E.), the Howard Hughes Medical Insitute (M.E. and J.R.E.), the NSF (IOS 1547796 and 1339125 to D.J.K.) the USDA National Institute of Food and Agriculture, (Hatch project number CA-D-PLS-7033-H to D.J.K.) and by the Danish National Research Foundation (DNRF99) grant.

## Author contributions
M.S. and M.E. conceived of the project. M.S. prepared samples. M.T. analyzed RNAseq data. B.L. and E.K. performed glucosinolate assays. M.S. generated transgenic lines, performed genetic experiments, and performed physiological experiments. D.J.K. assisted with glucosinolate analysis. M.S. and M.E. wrote the manuscript. M.S., M.E., L.S., J.R.E., E.K. and D.J.K. edited the manuscript. J.R.E. and L.S. provided materials.

## Additional information

**Competing interests:** The authors declare no competing interests.

