## [Peer Review File · Nature Communications]

Reviewer #1 (Remarks to the Author):

In this manuscript, Salehin et al. explore the role of three Aux/IAA Arabidopsis proteins (IAA5, IAA6 and IAA19) in controlling drought resistance via glucosinolate (GLS) level regulation. In previous work by the same group it was shown that these Aux/IAA genes are involved in drought tolerance and direct upstream regulators were described. Here they characterize the downstream events by utilizing triple recessive *iaa5 iaa6 iaa19* mutants. By RNA-seq they identified 12 genes that function in GLS biosynthesis that are downregulated in the triple mutant. They quantified the levels of GLS in the triple mutant and found one GLS (4-MSO) significantly downregulated. They tested mutations in GLS biosynthetic genes (*CYP79F1* and *F2*, *CYP83A1*) and found that these mutants show decreased drought tolerance. These genes are regulated by two TFs, MYB28 and 29, and discovered that the double *myb28 myb29* mutant is less drought tolerant. Conversely, overexpression lines show increased drought tolerance. By ChIP-PCR and RNA-seq they identified another TF WRKY63 as a direct target of IAA19 which in turn represses both MYB gene expression. Overexpression of WRKY63 results in decreased drought tolerance. Then they investigate whether this pathway affects stomata closure by testing mutants in Aux/IAA and MYB genes and treatments with 4-MSO and determined that failure to close stomata is responsible for decreased drought tolerance. They further determined that this pathway is independent from the plant hormone ABA role in stomata regulation and that it likely involves ROS and the receptor kinase GRH1 since the mutant *grh1* is resistant to 4-MSO applications. Finally, they determined that mutants in the auxin co-receptors AFB family results in increased tolerance to drought.

Understanding the molecular mechanisms regulating tolerance to drought stress is essential if we want to improve crops tolerance to extreme temperatures, so the topic of this manuscript is of broad significance. The manuscript in general is clear, concise and well-written, and the main conclusions are justified.

I have a few comments and suggestions below:

1. It is unclear whether the recombiner IAA19 line used for ChIP-PCR rescues the phenotype of the triple mutant. If this was shown in previous publication it should be mentioned, otherwise this data should be presented.
2. Fig. 1B, what is the scale in the heatmap?
3. There is quite a bit of variability in rosette dry weight quantifications in different experiments; see Col-O in Fig 2B-E for example (from 0.35-0.6). Is this normal?
4. In Fig.1 A, 13 genes are highlighted as differentially regulated but 12 are mentioned in the text.
5. *CYP79F2* is not down-regulated in the triple *iaa5 iaa6 iaa19* mutant but *cyp79f2* mutant does show less resistance to drought stress and they are both regulated by MYB28 and 29. How do you account for this difference given that in the triple mutant both MYBs are downregulated?
6. There is no mention of Fig. S4 in the text. I would also recommend the authors to include their findings on WRKY/MYB in the model.
7. Page 2: " We confirmed some of these results by..."
8. Page 3: introduce the AOP2 gene.
9. Page 4. Describe the *abi1-1* mutant when first introduced in the text.
10. Please specify how many bio-replicates were used in qRT-PCR experiments.
11. Fig 4C, the scale bar is over the text; please fix.
12. Page 3: "...further support for the idea that GLSs are..."
13. Page 4: introduce myrosinase TGG1.

Reviewer #2 (Remarks to the Author):

Glucosinolates (GLSs) are secondary metabolites that serve under biotic stress challenges. Here authors show that in Arabidopsis, aliphatic GLSs can additionally play a role in abiotic stress

response as they are stimulated by the auxin sensitive Aux/IAA repressors IAA5, IAA6, and IAA19 under drought conditions. Loss of IAA5/6/19 results in reduced GLS levels and decreased drought tolerance, which is further associated with a defect in stomatal regulation. Remarkably, authors recorded a complementation of stomatal closure by overexpression of MYBs regulating aliphatic GSLs and by exogenous application of aliphatic GSL 4MSO in IAA5/6/19 as well as myb28/myb29 mutants. Although it was not possible to explain what makes these metabolites so special in regulating stomatal closure or to track the direct regulator of aliphatic GSL biosynthesis and the direct regulator(s) of improved drought tolerance downstream of IAA5/6/19, this work is still a highlight. It shows that secondary metabolites can have functions beyond "secondary metabolic pathways" by attributing novel features of aliphatic GSL in drought stress. Series of experiments are comprehensive and results presented solid. I assume that findings of this work will melt borders of secondary and primary metabolism and help addressing improvement of plants growth under drought stress conditions.

Here are some of the questions I would like to be answered before publication:

- What makes aliphatic GSLs so special in their ability to regulate e.g. stomatal closure?
- Are aliphatic GSLs or ITCs the active molecules targeting essential component in plant cells?
- What are potential targets of these metabolic in plant cells vs human cell?
- How 4MSO is used to change stomata aperture in cells? Is active transport from vacuole or S-cells is needed?
- Do other metabolites will take over the function of aliphatic GSL in species containing no GSLs?
- Mutual negative regulation of indolic and aliphatic GSL pathways is known to take place upon overproduction of one of these two classes of metabolites. Could it be that plants measure levels of indolic GSLs produced by monitoring levels of IAA hormone and downregulating production of aliphatic GSL as a response to high IAA levels?
- A schematic drawing which shows common biosynthetic origin of IAA metabolite and indolic GSL along with mutual reciprocal negative regulation of indolic and aliphatic branches will be immensely helpful (in Supplemental Data).
- This reviewer also wonders whether mechanism ascribed to aliphatic GSLs are as GSL causative as proposed. Figure 2 and Figure 4 present very impressive results suggesting this should be the case. But, is it possible that MYB28 and MYB29 have target genes beyond GSL and related to drought response? Public microarray or RNA-seq data? Furthermore, could it be that treatment of Arabidopsis plants with 4MSO induce expression of MYB28 or 29 followed by activation of non-GSL genes. Addressing/discussing a potential of having non GSL target genes in the arsenal of MYB28/29 regulators would help unveiling this doubt.

Minor points:

- For some but not all experiments authors used the *cyp79f1/f2* mutants. Need to be mentioned why is this so and whether authors observed the phenotypic features of *cyp79f1/f2* known as "bushy". Do *cyp79f1/f2* plants have different IAA levels?
- Data set presented in Figure 2 (complementation of drought stress phenotype of IAA19 mutant by overexpression of MYBs regulating GSLs) is very impressive and very nice. Is there a reason why authors used a single *iaa19* mutant and not *iaa5/6/19* mutant.
- Complementation of *iaa19* with MYB34 or 51 would be of interest. It can eventually address the potential of indolic MYBs (and ancestor genes of MYB28 and MYB29) having ability to activate genes beyond GSLs.

Reviewer #3 (Remarks to the Author):

The authors clarified that aliphatic glucosinolate (GLS) levels are regulated by the auxin-sensitive Aux/IAA repressors IAA5, IAA6, and IAA19 using Arabidopsis. These proteins act in a transcriptional cascade that maintains expression of GLS levels in plants under drought conditions. Loss of IAA5, IAA6, and IAA19 reduced GLS levels and decreased drought tolerance through stomatal regulation. In addition, application of GLS to the *iaa5 iaa6 iaa19* mutants restores

stomatal regulation and normal drought tolerance. The authors claimed GLS action is dependent on the receptor kinase GHR1. This topic is very interesting but there are several points to be addressed and to be improved.

The authors propose the model (Fig. S4.). In this model, myrosinase mediates the reaction from GLS to ITC. Where is the myrosinase present? Also where is the GLS?

The authors tested effects of exogenous GLS on stomatal responses? In this case, can GLS be contacted with myrosinase?

In addition, the authors should provide information about solvents for 4-MSO and I3M.

Islam et al. reported that ITC inhibited potassium inward-rectifying channels. The authors should cite this paper and discuss their results because they examined light-induced stomatal opening, which is strongly regulated by the potassium channels.

The authors should measure ITC contents if proposing the model.

Response to reviewers:

We are very grateful to the reviewers for their many insightful comments. We have tried to address each of the reviewers concerns as described below. As we considered the revised manuscript, we also realized that we had used the wrong wild-type control (Col-o instead of Ws) for experiments involving the *cyp79f1f2* double mutant. We have repeated these experiments using the Ws control line.

Mark Estelle

Reviewers' comments:

Reviewer #1 (Remarks to the Author):

In this manuscript, Salehin et al. explore the role of three Aux/IAA Arabidopsis proteins (IAA5, IAA6 and IAA19) in controlling drought resistance via glucosinolate (GLS) level regulation. In previous work by the same group it was shown that these Aux/IAA genes are involved in drought tolerance and direct upstream regulators were described. Here they characterize the downstream events by utilizing triple recessive *iaa5 iaa6 iaa19* mutants. By RNA-seq they identified 12 genes that function in GLS biosynthesis that are downregulated in the triple mutant. They quantified the levels of GLS in the triple mutant and found one GLS (4-MSO) significantly downregulated. They tested mutations in GLS biosynthetic genes (*CYP79F1* and *F2*, *CYP83A1*) and found that these mutants show decreased drought tolerance. These genes are regulated by two TFs, MYB28 and 29, and discovered that the double *myb28 myb29* mutant is less drought tolerant. Conversely, overexpression lines show increased drought tolerance. By ChIP-PCR and RNA-seq they identified another TF WRKY63 as a direct target of IAA19 which in turn represses both MYB gene expression. Overexpression of WRKY63 results in decreased drought tolerance. Then they investigate whether this pathway affects stomata closure by testing mutants in Aux/IAA and MYB genes and treatments with 4-MSO and determined that failure to close stomata is responsible for decreased drought tolerance. They further determined that this pathway is independent from the plant hormone ABA role in stomata regulation and that it likely involves ROS and the receptor kinase GRH1 since the mutant *grsh1* is resistant to 4-MSO applications. Finally, they determined that mutants in the auxin co-receptors AFB family results in increased tolerance to drought.

Understanding the molecular mechanisms regulating tolerance to drought stress is essential if we want to improve crops tolerance to extreme temperatures, so the topic of this manuscript is of broad significance. The manuscript in general is clear, concise and well-written, and the main conclusions are justified.

I have a few comments and suggestions below:

1. It is unclear whether the recombineered IAA19 line used for ChIP-PCR rescues the phenotype of the triple mutant. If this was shown in previous publication it should be mentioned, otherwise this data should be presented.

*Yes, the characterization of that line, including its ability to rescue the *iaa19* mutant, was presented in Shani et al., 2017.*

2. Fig. 1B, what is the scale in the heatmap?

Our apologies. The scale disappeared in the PDF.

3. There is quite a bit of variability in rosette dry weight quantifications in different experiments; see Col-O in Fig 2B-E for example (from 0.35-0.6). Is this normal?

We have noted this as well. We think this is due to seasonal differences in humidity. Our chambers don't have humidity regulation. It is important to note that each experiment was repeated three times and the same trend was observed in each case.

4. In Fig.1 A, 13 genes are highlighted as differentially regulated but 12 are mentioned in the text.

Corrected.

5. CYP79F2 is not down-regulated in the triple *iaa5 iaa6 iaa19* mutant but *cyp79f2* mutant does show less resistance to drought stress and they are both regulated by MYB28 and 29. How do you account for this difference given that in the triple mutant both MYBs are downregulated?

CYP79F2 was downregulated in the RNAseq data but with a FDR of 0.049, it was below our FDR threshold of 0.001. Nevertheless, we assayed CYP79F2 expression by qRT-PCR. The results in Fig S1B show that this gene is downregulated in the mutant, although not as much as some of the other genes in the pathway.

6. There is no mention of Fig. S4 in the text. I would also recommend the authors to include their findings on WRKY/MYB in the model.

Corrected.

7. Page 2: " We confirmed some of these results by..."

Corrected

8. Page 3: introduce the AOP2 gene.

Done.

9. Page 4. Describe the *abi1-1* mutant when first introduced in the text.

Done.

10. Please specify how many bio-replicates were used in qRT-PCR experiments.

All qRT-PCR experiments had 3 bioreplicates. We now indicate this in the Methods.

11. Fig 4C, the scale bar is over the text; please fix.

Corrected.

12. Page 3: "...further support for the idea that GLSs are..."

Corrected.

13. Page 4: introduce myrosinase TGG1.

The myrosinase is already introduced on page 1.

Reviewer #2 (Remarks to the Author):

Glucosinolates (GLSs) are secondary metabolites that serve under biotic stress challenges. Here authors show that in Arabidopsis, aliphatic GLSs can additionally play a role in abiotic stress response as they are stimulated by the auxin sensitive Aux/IAA repressors IAA5, IAA6, and IAA19 under drought conditions. Loss of IAA5/6/19 results in reduced GLS levels and decreased drought tolerance, which is further associated with a defect in stomatal regulation. Remarkably, authors recorded a complementation of stomatal closure by overexpression of MYBs regulating aliphatic GSLs and by exogenous application of aliphatic GSL 4MSO in IAA5/6/19 as well as myb28/myb29 mutants. Although it was not possible to explain what makes these metabolites so special in regulating stomatal closure or to track the direct regulator of aliphatic GSL biosynthesis and the direct regulator(s) of improved drought tolerance downstream of IAA5/6/19, this work is still a highlight. It shows that secondary metabolites can have functions beyond "secondary metabolic pathways" by attributing novel features of aliphatic GSL in drought stress. Series of experiments are comprehensive and results presented solid. I assume that findings of this work will melt borders of secondary and primary metabolism and help addressing improvement of plants growth under drought stress conditions.

Here are some of the questions I would like to be answered before publication:

- What makes aliphatic GSLs so special in their ability to regulate e.g. stomatal closure?

At this point we don't have an answer to this important question. Because we have tested a small number of compounds, we hesitate to draw strong conclusions. We expect that specificity is related to the interaction between the ITC, and its target, perhaps a peroxidase. These intriguing questions await further experimentation.

- Are aliphatic GSLs or ITCs the active molecules targeting essential component in plant cells?

Based on Sally Assmann's work (Zhao et al., 2008), ITCs appear to be the active molecule. We now state this more directly on page 4. We have also shown the tgg1 tgg2 double mutant is

sensitive to growth on medium containing PEG indicating that a myrosinase dependent breakdown product of the GLS is required for stress tolerance. We now include this data in Fig S3C.

- What are potential targets of these metabolic in plant cells vs human cell?

Our understanding is that the targets of glucosinolates in human cells are largely unknown. In any case, I don't think our study speaks to this at present. Perhaps in the future. In plants, the best-known targets of glucosinolates are insect foragers and some fungal and bacterial pathogens, although even in these cases, the molecular targets are not well understood. With respect to our work, based on earlier studies we propose that the ITCs promote ROS formation through a peroxidase. A confirmation of this model awaits further experimentation.

- How 4MSO is used to change stomata aperture in cells? Is active transport from vacuole or S-cells is needed?

This is an interesting topic that we don't address in our paper. The source of the GLS that regulates the stomata is currently unknown, although we have preliminary data that addresses this question. We prefer to hold this data for another manuscript.

- Do other metabolites will take over the function of aliphatic GSL in species containing no GSLs?

*This is an excellent question. Unfortunately, we don't know the answer. Previous work has shown that ITCs regulate stomata in *Vicia faba* and it is possible that endogenous ITC is produced through a different pathway in this and other plants. We now mention this speculation on page 6.*

- Mutual negative regulation of indolic and aliphatic GSL pathways is known to take place upon overproduction of one of these two classes of metabolites. Could it be that plants measure levels of indolic GSLs produced by monitoring levels of IAA hormone and downregulating production of aliphatic GSL as a response to high IAA levels?

This is a very interesting hypothesis that we, or perhaps others, can test in the future.

- A schematic drawing which shows common biosynthetic origin of IAA metabolite and indolic GSL along with mutual reciprocal negative regulation of indolic and aliphatic branches will be immensely helpful (in Supplemental Data).

We agree that these are very interesting questions, and no doubt the regulation of glucosinolate biosynthesis and transport is complex. However, we are not convinced that such a figure would be helpful given the more focused nature of our study. We would prefer not to include this unless the reviewer insists.

- This reviewer also wonders whether mechanism ascribed to aliphatic GSLs are as GSL causative as proposed. Figure 2 and Figure 4 present very impressive results suggesting this

should be the case. But, is it possible that MYB28 and MYB29 have target genes beyond GSL and related to drought response? Public microarray or RNA-seq data? Furthermore, could it be that treatment of Arabidopsis plants with 4MSO induce expression of MYB28 or 29 followed by activation of non-GSL genes. Addressing/discussing a potential of having non GSL target genes in the arsenal of MYB28/29 regulators would help unveiling this doubt.

Yes, we considered this possibility. However, because over-expression of AOP2, a downstream enzyme in GLS biosynthesis, also results in drought tolerance, we don't think other MYB28/29 targets are likely to be causal.

Minor points:

- For some but not all experiments authors used the *cyp79f1/f2* mutants. Need to be mentioned why is this so and whether authors observed the phenotypic features of *cyp79f1/f2* known as "bushy". Do *cyp79f1/f2* plants have different IAA levels?

*Both the *cyp79f1/f2* and the *myb28 myb29* lines have very low levels of GLS. We chose to work with the latter because it is representative.*

*The *cyp79f1f2* double mutant does exhibit a bushy phenotype in our hands as has been described, and it has also been reported that these plants exhibit higher auxin levels (Tantikanjana et al., Genes and Dev., 2001). However, we think that it is unlikely that the change in auxin levels results in decreased tolerance since none of the other genotypes we tested (*cyp83a1*, *myb28/29*, *35S:MYB28* etc) exhibit an auxin related phenotype. Also, the fact that directly application of GLS can rescue the drought phenotype strongly argues that a change in GLS levels is responsible for this phenotype.*

- Data set presented in Figure 2 (complementation of drought stress phenotype of IAA19 mutant by overexpression of MYBs regulating GSLs) is very impressive and very nice. Is there a reason why authors used a single *iaa19* mutant and not *iaa5/6/19* mutant.

*We chose the single mutant because it is easier to introgress that *35S:MYBs* into this line than the triple mutant and the single has a similar phenotype to the triple mutant.*

- Complementation of *iaa19* with MYB34 or 51 would be of interest. It can eventually address the potential of indolic MYBs (and ancestor genes of MYB28 and MYB29) having ability to activate genes beyond GSLs.

Another good suggestion for the future.

Reviewer #3 (Remarks to the Author):

The authors clarified that aliphatic glucosinolate (GLS) levels are regulated by the auxin-sensitive Aux/IAA repressors IAA5, IAA6, and IAA19 using Arabidopsis. These proteins act in a transcriptional cascade that maintains expression of GLS levels in plants under drought conditions. Loss of IAA5, IAA6, and IAA19 reduced GLS levels and decreased drought tolerance through stomatal regulation. In addition, application of GLS to the *iaa5 iaa6 iaa19* mutants

restores stomatal regulation and normal drought tolerance. The authors claimed GLS action is dependent on the receptor kinase GHR1. This topic is very interesting but there are several points to be addressed and to be improved.

The authors propose the model (Fig. S4.). In this model, myrosinase mediates the reaction from GLS to ITC. Where is the myrosinase present? Also where is the GLS?

As we mention in the manuscript, previous studies have shown that the guard cells have high levels of TGG1. However, at this point we don't the location of the GLS. We are working to determine if GLS is transported from out of the GC but this is work for another manuscript.

The authors tested effects of exogenous GLS on stomatal responses? In this case, can GLS be contacted with myrosinase?

At present we don't know the answer to this question.

In addition, the authors should provide information about solvents for 4-MSO and I3M.

4-MSO was dissolved in water while I3M was dissolved in DMSO.

Islam et al. reported that ITC inhibited potassium inward-rectifying channels. The authors should cite this paper and discuss their results because they examined light-induced stomatal opening, which is strongly regulated by the potassium channels.

We were unable to find the manuscript that the reviewer refers to. Islam et al published a paper on the effects of acrolein on stomatal regulation, but since acrolein is not structurally related to ITC, we are not sure that it makes sense to describe this work in our manuscript. However, we acknowledge that the precise mechanism of GLS action is far from clear and have added a statement to this effect to our manuscript.

The authors should measure ITC contents if proposing the model.

We agree that this is the logical next step. We are now developing methods to measure GLS and ITC at cellular resolution. In the absence of this data, we think it is still valuable to present a model that can explain our results.

REVIEWERS' COMMENTS:

Reviewer #1 (Remarks to the Author):

I had reviewed a previous version of this manuscript and all of my questions and concerns were appropriately answered. This manuscript uncovers the molecular mechanisms behind the connection between auxin and secondary metabolism in regulating plant growth in response to drought stress. The results presented are novel and will stimulate further research on the topic. The manuscript is clear and the main conclusions are justified.

Reviewer #2 (Remarks to the Author):

Authors addressed most of the reviewers concerns very well. They substantially revised this work. However this reviewer still three remaining concerns:

(i) mechanism behind the central message of the ms, which is about GLS-mediated drought tolerance, remain unknown.

(ii) it remains unclear how secondary metabolites known for their role in biotic defense can affect drought tolerance or stomata closure. E.g. It is still puzzling how GSLs or their degradation products are transported to the sites of their activity in stomata. The steps from GSL storage place, to the transport, to the activation of myrosinase, generating ITCs and interaction with peroxidase in Figure 5 remains unexplained.

(iii) How important is this specific drought resistance pathway in plants accumulating no GLS or ITC?

Measurement of the potential active molecule and probably the inducer of drought activity will be of much help.

Reviewer #3 (Remarks to the Author):

The authors revised the manuscript according to the reviewers' comments but not sufficiently. Their model is not totally supported by their results and several essential data are missing including their localization and ITC contents.

It is unclear that the exogenous GLSs are converted to ITCs by myrosinases. ITCs content data are necessary.

The authors dissolved 4-MSO in water and I3M in DMSO and probably ABA in ethanol. Figure 4B, 4C, and 4C: What is "control" for each? Does it contain ethanol or DMSO?

Response to Reviewers

Dear Reviewers. Thanks once again for your very helpful comments.

Reviewer #1

Thanks for your kind comments.

Reviewer #2

However this reviewer still three remaining concerns:

(i) mechanism behind the central message of the ms, which is about GLS-mediated drought tolerance, remain unknown.

(ii) it remains unclear how secondary metabolites known for their role in biotic defense can affect drought tolerance or stomata closure. E.g. It is still puzzling how GSLs or their degradation products are transported to the sites of their activity in stomata. The steps from GSL storage place, to the transport, to the activation of myrosinase, generating ITCs and interaction with peroxidase in Figure 5 remains unexplained.

(iii) How important is this specific drought resistance pathway in plants accumulating no GLS or ITC?

Measurement of the potential active molecule and probably the inducer of drought activity will be of much help.

We agree with this reviewer that there are many outstanding questions that require further study. However, we think that the core conclusions from our paper, that auxin regulates GLS biosynthesis and that GLS regulation of stomatal aperture to promote drought tolerance, are major advances in the field. Now that we know that GLS compounds are required for drought tolerance, we can focus on downstream events.

Reviewer #3

The authors revised the manuscript according to the reviewers' comments but not sufficiently. Their model is not totally supported by their results and several essential data are missing including their localization and ITC contents.

It is unclear that the exogenous GLSs are converted to ITCs by myrosinases. ITCs content data are necessary.

We agree that we may have focused too much on ITC as the active GLS breakdown product. Although we think that identification of the molecule is beyond the scope of this study, we have modified the text and Figure 5 to acknowledge that other GLS breakdown products may be involved.

- *On page 3, first paragraph we state that “**GLSs are broken down by the enzyme myrosinase into thiohydroximate-O-sulfonates which rearrange to form diverse isothiocyanates (ITC), nitriles and related compounds (8-12)**”.*

- On page 5, first paragraph we have changed the text to read **“Together these results indicate that myrosinase-dependent breakdown products of applied GLS promote stomatal closure.”** modifying the focus on ITCs
- In the Discussion, page 6 and 7, we change text to read **“GLS degradation products, perhaps ITCs but possible other compounds, stimulate the formation of ROS”**. Later we state that **“Whether or not there is a source of endogenous ITC or related molecule in beans and other non- Brassicaceae remains to be determined”**.
- In Figure 5, we have replaced the solid lines leading from GLS to ROS with dotted lines. In addition, we placed a question mark next to ITC and changed the text in the legend slightly to read **“Myrosinase acts on GLS to produce degradation products, including isothiocyanates (ITC), that promote ROS production via a peroxidase”**

The authors dissolved 4-MSO in water and I3M in DMSO and probably ABA in ethanol. Figure 4B, 4C, and 4C: What is “control” for each? Does it contain ethanol or DMSO?

ABA was dissolved in ethanol. For stomatal aperture experiments, the appropriate amount of DMSO (I3M) and ethanol was added to the control solution. We now note this in the Materials and Methods